# Estimating Generic 3D Room Structures from 2D Annotations

Denys Rozumnyi[2]*    Stefan Popov[1]    Kevis-Kokitsi Maninis[1]
Matthias Nießner[3]    Vittorio Ferrari[1]

[1]Google Research    [2]ETH Zurich    [3]Technical University of Munich

## Abstract

Indoor rooms are among the most common use cases in 3D scene understanding. Current state-of-the-art methods for this task are driven by large annotated datasets. Room layouts are especially important, consisting of structural elements in 3D, such as wall, floor, and ceiling. However, they are difficult to annotate, especially on pure RGB video. We propose a novel method to produce generic 3D room layouts just from 2D segmentation masks, which are easy to annotate for humans. Based on these 2D annotations, we automatically reconstruct 3D plane equations for the structural elements and their spatial extent in the scene, and connect adjacent elements at the appropriate contact edges. We annotate and publicly release 2246 3D room layouts on the RealEstate10k dataset, containing YouTube videos. We demonstrate the high quality of these 3D layouts annotations with extensive experiments.

## 1 Introduction

Estimating 3D structures from video is one of the key tasks in computer vision. Many methods have been proposed, including Structure-from-Motion [49] and Multi-View Stereo [50, 15] pipelines. In particular, indoor room furniture [36, 43] and layout [4, 41, 65, 56] estimation is becoming more important in recent years.

The key ingredient of state-of-the-art methods are large annotated datasets for training. A variety of real [7, 57, 3] and synthetic [72, 16, 27] datasets with annotated 3D objects are available. Some datasets even offer 3D room layout, which is defined as a set of 3D structural elements such as wall, floor, and ceiling. Synthetic datasets [72, 16, 27] typically provide 3D room layouts along with 3D objects. In contrast, most real datasets focus only on 3D object annotations [7, 55, 57, 44, 2, 62]. There are just a few real datasets with annotated 3D room layouts. However, they require the images/videos to be acquired with special devices or sensors such as RGB-D cameras or panoramic captures [4, 55, 77, 76, 10, 70, 65]. This limits the range of scenes they can cover. Moreover, these datasets often offer only simple room structures such as Cuboid [70, 76, 19, 66] or Manhattan [65, 77].

In this paper we propose a method for annotating generic 3D room layouts on commonly-available RGB videos. A naive system for annotating such 3D room layouts would ask human annotators to draw layouts in 3D. However, this is an extremely hard task, which would require very high abstraction and 3D representation abilities from the annotators, as well as a very advanced user interface. Instead, we propose a method to annotate 3D room layouts that is much easier for human annotators: draw an amodal segmentation mask for each structural element *in 2D*, and also approximately mark its visible parts *in 2D* (Fig. 1). This reduces the sophisticated target 3D task to simply drawing segmentation masks in 2D, which is a common annotation process used to create 2D datasets [39, 32, 74]. Moreover,

---

*Work done while the author was at Google.

37th Conference on Neural Information Processing Systems (NeurIPS 2023) Track on Datasets and Benchmarks.

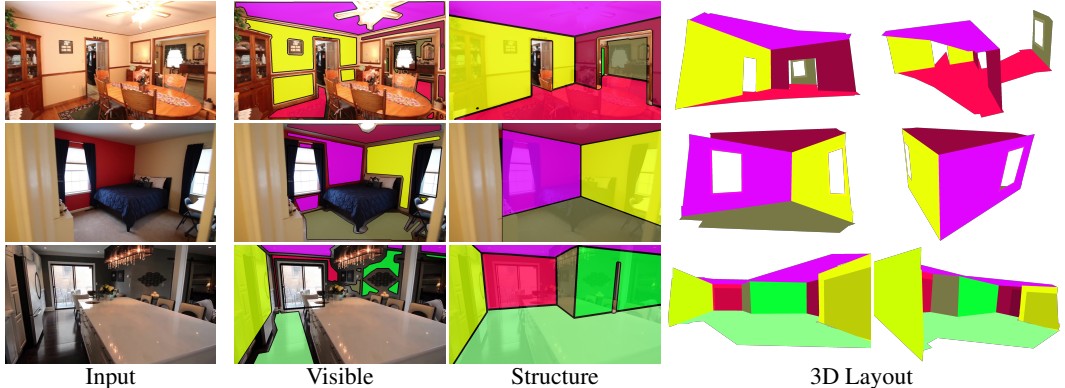

| Input | Visible | Structure | 3D Layout |
|-------|---------|-----------|-----------|

Figure 1: **From 2D annotations to 3D room layouts.** Given an RGB video, we ask human annotators to draw amodal segmentation masks and visible parts of each structural element, *e.g.* wall, floor. Then, our method automatically derives high-quality 3D layouts from these 2D annotations.

all annotation is performed on each video frame *independently* without requiring the annotator to provide correspondences among video frames.

The simplification of the annotator task is enabled by shifting much of the work to an automatic method we propose, capable to derive 3D room layouts from these 2D annotations. This method estimates a 3D plane equation for each structural element, as well as a finite spatial extent that captures all parts of the plane that are in the camera's field-of-view at any time during the video. We estimate all elements jointly, and connect adjacent elements at the right contact edges in 3D, matching their shared edge as observed in the 2D annotations.

Using the proposed approach, we annotate 2246 scenes from the RealEstate10k [75] dataset that contains YouTube videos of indoor scenes. The rooms are complex and cover generic types, not limited to Cuboid/Manhattan. They can even be composite, such as two rooms connected by a door or a staircase. The method also works when the video does not show the full room (a common case). We evaluate the quality of the produced 3D layout annotations on RealEstate10k in terms of Reprojection Intersection-over-Union (IoU) to the input ground-truth 2D segmentation masks, and on ScanNet [11], which enables measuring 3D depth errors as it features ground-truth depth maps. We find the 3D layouts to be highly accurate, with only about 20cm depth error (out of 7 meter wide scenes), and high Reprojection IoU around 0.9. We also manually inspected the 3D layouts, finding that we reconstruct correctly about 98% of all structural elements that are actually in the scene. We have released the dataset publicly at `https://github.com/google-research/cad-estate`.

## 2 Related work

**Datasets of 3D indoor scenes.** Many datasets with 3D indoor scenes were proposed (see Table 1). Synthetic datasets have ground-truth 3D room layouts by construction, *e.g.* Structured3D [72], 3D-FRONT [16, 17], InteriorNet [27], House3D [60], SUNCG [53], SceneNet [38], OpenRooms [28]. However, the imagery is not real, causing a domain gap when training models on them [59, 22, 67, 46].

Many datasets of real images have only 3D objects annotated, without structural elements (see Table 1). A few real datasets with 3D room layouts were introduced, but they heavily rely on special acquisition devices or sensors to create the 3D annotation of the scene, which limits the range of scenes they can cover. For instance, Zillow Indoor Dataset [10], Realtor360 [65], and PanoContext [70] annotate layouts on 360° panoramic images. SceneCAD [4] was acquired with an RGB-D sensor. MatterportLayout [77] and LayoutNet [76] require both a depth sensor and panoramic images.

Most of these datasets cover rooms with simple layouts, that are either: (a) Cuboid [70, 76, 19, 66], or (b) Manhattan [65, 77] (all structural elements are aligned with a canonical coordinate system), and the methods to construct them do not work in more complex room structures. Some datasets [10, 4] construct generic room layouts (like ours), but are limited by the data capturing process (see above).

| Dataset | Data type | #Scenes | #Images | Objects | Layout | Type |
|---|---|---|---|---|---|---|
| [72, 16, 27, 60, 53, 38] | synthetic | varies | varies | ✓ | ✓ | G |
| [2, 44, 57, 9, 29] | RGB | varies | varies | ✓ | - | - |
| [21, 61, 62, 52, 3] | RGB-D | varies | varies | ✓ | - | - |
| Replica [55] | RGB-D video | 18 | - | ✓ | - | G |
| SceneCAD [4] | RGB-D video | 1,151 | - | ✓ | ✓ | G |
| MatterportLayout [77] | RGB-D pano image | 2,295 | 2,295 | - | ✓ | M |
| LayoutNet [76] | RGB-D pano image | 571 | 571 | - | ✓ | C |
| Zillow Indoor [10] | RGB pano video | 1,524 | 71,474 | - | ✓ | G |
| Realtor360 [65] | RGB pano image | 2,573 | 2,573 | - | ✓ | M |
| PanoContext [70] | RGB pano image | 700 | 700 | - | ✓ | C |
| Ours | RGB video | 2,246 | 375,677 | - | ✓ | G |

Table 1: **Related datasets with 3D room structures.** We distinguish layout types: Cuboid (C), Manhattan (M), Generic (G). We propose the first dataset with generic 3D room layouts that were annotated without any special acquisition devices or sensors and only from an RGB video.

In contrast to the above works, we propose a method for annotating generic 3D room layouts on commonly-available RGB videos. Moreover, our annotation process is easy for human annotators, involving only drawing in 2D. With it, we annotate 2246 RGB videos from the RealEstate10k dataset [75], which we release publicly.

**Layout estimation methods.** A variety of methods for 2D/3D room layout estimation have been proposed. Some methods estimate from dense 3D point clouds derived from RGB-D fusion [40, 4, 8, 18] and some from a single panoramic RGB image [41, 65, 56, 63, 6, 76]. Also, many methods estimate a 2D room layout (*i.e.* 2D projection of a 3D room layout) from a single RGB image [25, 31, 73, 69, 54, 19, 20, 13, 14, 34, 33, 64]. They are trained on datasets with such 2D room layouts annotations [19, 66]. Most methods make some additional assumptions such as Cuboid [25, 31, 73, 19, 20] or Manhattan [40] rooms. This family of methods could benefit from our new dataset.

**Machine-assisted annotation.** Many datasets in computer vision have been generated using machine-assisted annotations. Among them are object segmentation with bounding boxes [47, 26], scribbles [30, 1, 5], extreme points [37], language models [48], and even localized captions [42]. We believe ours is the first method for machine-assisted 3D room layout annotation.

## 3 Method

We want to estimate the layout of the room in an input RGB video. A layout is a set of structural elements in 3D, each with an instance index from one of the following classes: Floor, Ceiling, Wall, Slanted (could be a wall, ceiling, *etc.*), Door, and Window. Each structural element is represented by a 3D plane with a finite spatial extent capturing all parts that are in the camera's field-of-view at any time during the video. This spatial extent is independent of room furniture, which means structural elements should be estimated even behind it. Finally, the spatial extents of neighboring elements are expected to be connected at the right contact edges in 3D.

Our method inputs video frames $\{I_1, \ldots, I_N\}$, along with their camera intrinsics and extrinsics, *e.g.* computed by COLMAP [49]. The main outputs are the 3D plane equations $\{p_1, \ldots, p_M\}$ of each structural element and a triangular 3D mesh denoting their spatial extent.

In the annotation stage, annotators draw an amodal segmentation mask for each structural element *in 2D*, and also approximately mark its visible parts *in 2D*, for each frame *independently* from a small subset of input frames (every 30th, Sec. 3.1). Then, we propose an automatic pipeline to generate 3D room layouts from these 2D annotations (Fig. 2). First, we track 2D points on visible parts and assign point tracks to structural elements (Sec. 3.2). Then, we estimate a 3D plane equation for each structural element by minimizing a joint loss function that combines terms for fitting the 3D geometry induced by its associated point tracks, matching the edges between structural elements marked in the 2D annotations, and encouraging vertical walls to be perpendicular to the floor and ceiling (Sec. 3.3). Next, we estimate the spatial extent of each plane as a union of the associated

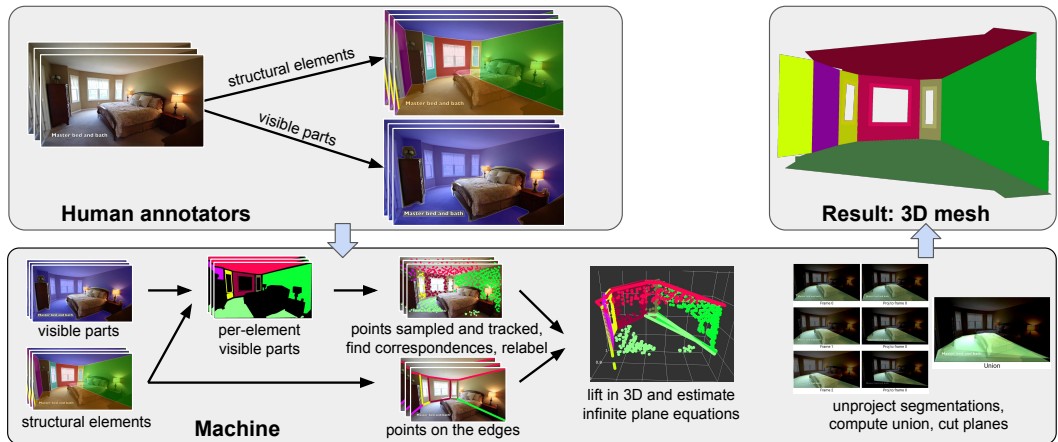

Figure 2: **Pipeline overview.** For each selected video frame independently, humans annotate segmentation masks for structural elements and their visible parts. We track 2D points on visible parts and use the 3D geometry induced by them to estimate 3D plane equations for the structural elements. Finally, we estimate the spatial extent of each structural element as a union of unprojected 2D annotations across all frames.

plane's amodal segmentation over all frames (Fig. 2, bottom box, rightmost; Sec. 3.4) ). Finally, we automatically measure reconstruction quality, and we retain only videos where it is high (Sec. 3.5).

### 3.1   Human 2D annotations

**2D Structural elements annotation.**   Annotators draw an amodal segmentation mask for each structural element as a polygon, *i.e.* including its entire surface even if occluded by furniture (covering classes Floor, Ceiling, Wall, Slanted, Door, Window; Fig. 2). Moreover, we also ask annotators to draw *occlusion edges*. An occlusion edge is a curve of arbitrarily shape separating two structural elements that do not meet in 3D along this edge, *e.g.* when one wall occludes part of another wall behind it (Fig. 1, top row, occlusion edges between olive and dark red walls).

**2D Visible parts annotation.**   We additionally ask annotators to mark the visible parts of all structural elements approximately in selected frames (Fig. 2). Its main purpose is to guide the process of establishing correspondences between annotations across frames (Sec. 3.2), which in turn helps estimating 3D plane equations (Sec. 3.3). Annotators are not required to label all visible pixels accurately, but to focus on regions with non-uniform texture including grating, shadows, flat objects near the surface like posters, paintings, *etc*. As a rule of thumb, they should exclude objects protruding more than 3 cm from the plane surface, furniture, glossy surfaces (*e.g.* the glass frame of a picture), and artificially added overlays (*e.g.* captions).

### 3.2   2D Point tracking

A key component of our method is tracking 2D points sampled on the visible parts of structural elements. We sample points with low-discrepancy Sobol sampling [51], and then track them over time with RAFT [58]. This process results in many point tracks. A track $x_t$ spans multiple frames $f$, each with a 2D point $x_t^f$.

**Structural elements correspondences.**   Since video frames are annotated independently, labels of the same structural elements do not necessarily match across frames. We solve this problem by establishing correspondences based on the point tracks. For each pair of consecutive annotated frames, we create a correspondence matrix between each pair of structural elements they contain. An entry $(i, j)$ represents the number of tracked points that have label $i$ in the first frame and label $j$ in the second frame. Then, we compute the optimal matching between structural elements by the Hungarian method [24]. This way, the instance indices are consistent over time, with each index representing the same structural element across all video frames.

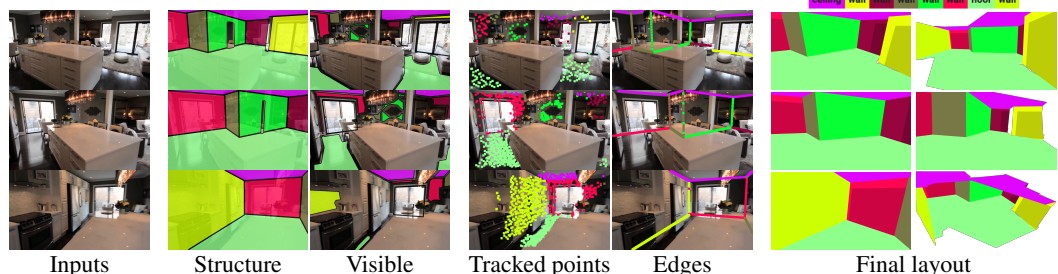

| Inputs | Structure | Visible | Tracked points | Edges | Final layout |

Figure 3: **3D room layout reconstruction on an example scene.** Given an input video and manual 2D annotations of structural elements and their visible parts, we combine point tracks fitting, edge matching, and perpendicularity constraints to generate a 3D room layout.

**Assigning 2D point tracks to structural elements.** The process just described also determines which structural element each point track belongs to. Formally, the process assigns a structural element index $j$ to each track $x_t$, which we denote by $a(t) = j$. This will be useful below, as we try to fit a 3D plane based on the evidence from the point tracks assigned to it.

### 3.3 Estimating 3D plane equations

Each structural element is represented by its 3D plane equation and spatial extent. The equation $p$ has 4 parameters representing the plane's 3D normal and offset from the origin. To estimate it, we optimize the parameters of all planes jointly, so as to minimize a loss involving three terms. First, the 3D plane should fit well the 3D geometry induced by the tracked points assigned to that structural element (*Fitting point tracks loss*). Second, the edges between structural elements in the 2D annotations should correspond to the intersections of their corresponding 3D planes (*Edge loss*). Third, vertical walls should be perpendicular to the floor and ceiling (*Perpendicularity loss*).

**Fitting point tracks loss.** Consider a 2D point track $x_t$ assigned to a structural element $j = a(t)$. Now also consider a candidate plane equation $p_j$ for that structural element, determining the plane in 3D space. We can now *unproject* a 2D point $x_t^f$ from the track into a straight line in 3D space, given the camera parameters for frame $f$ (Fig. 4). This line will intersect the candidate plane at a 3D point $X_t^f$. We denote this process of unprojection and plane intersection as $X_t^f = \text{unproj}(x_t^f, p_j, f)$, and refer to it as "unprojecting point $x_t^f$ onto plane $p_j$".

In practice, all 2D points $x_t^f$ within a single track $x_t$ correspond to the same point in 3D space. Hence, we define a loss that prefers a plane equation $p_j$ such that the 3D unprojection lines for each point within a point track intersect that plane at the same point:

$$\frac{1}{|x_t|} \sum_f \left\| \text{unproj}(x_t^f, p_j, f) - X_t \right\| \tag{1}$$

where $X_t = \frac{1}{|x_t|} \sum_f \text{unproj}(x_t^f, p_j, f)$ is the average over all unprojected points for the track; the sum over $f$ runs over all frames where track $x_t$ is visible; and $|x_t|$ is the number of such frames (*i.e.* the track length). Basically, the loss penalizes the distance of each frame's unprojections to their mean. The complete loss runs over all point tracks $t$:

$$\mathcal{L}_T = \frac{1}{T} \sum_{t=1}^{T} \frac{1}{|x_t|} \sum_f \left\| \text{unproj}(x_t^f, p_{a(t)}, f) - X_t \right\| \tag{2}$$

Note how this includes *all* point tracks that are assigned to various different planes, as marked in $a(t)$.

**Edge loss.** This loss encourages the intersection of two structural elements in 3D to project to the edge between their 2D annotations. In other words, an edge between two structural elements in the 2D annotations unprojects to the intersection of the corresponding 3D planes. The core constraint can be phrased formally as: when the same 2D point is unprojected onto two different 3D planes, the resulting 3D points will be equal only if the two 3D planes intersect at that 3D point.

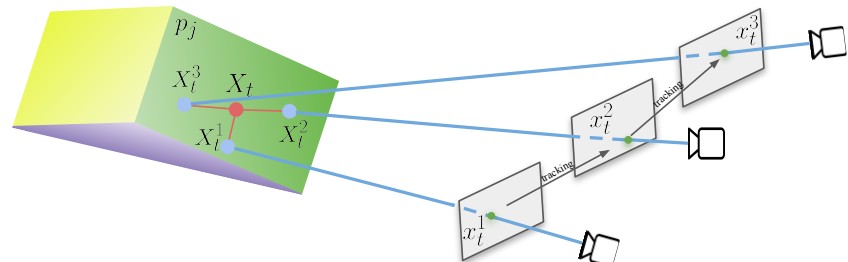

Figure 4: **Unprojecting 2D points onto a 3D plane.** Each point $x_t^f$ (green) in a point track is unprojected onto its assigned plane $p_j$ as a 3D point $X_t^f$ (blue). This 3D point is the intersection of the unprojected line (blue) and the 3D plane. If we had the ideal 3D plane, all these lines would intersect at the same 3D point. In practice, we fit a plane equation by minimizing a loss on the distance of $X_t^f$ to their mean $X_t$ (red).

To compute the loss, we first extract 2D edge points $e_k$ that lie at the boundary between two structural elements in an annotated frame. For each edge point $e_k$, we denote the frame where it lies by $f(k)$, and the indices of the two structural elements it connects by $i(k)$ and $j(k)$. It is defined as $\mathcal{L}_E =$

$$\frac{1}{E} \sum_{k=1}^{E} \left\| \text{unproj}\Big(e_k, p_{i(k)}, f(k)\Big) - \text{unproj}\Big(e_k, p_{j(k)}, f(k)\Big) \right\| \tag{3}$$

where $E$ is the total number of edge points across all frames and all structural elements.

**Perpendicularity loss.** Since we know that vertical walls are perpendicular to the floor and ceiling, we minimize the cosine of the angle between 3D planes associated to these structural elements (slanted structural elements are not affected by this loss). The set of pairs of Wall and Ceiling/Floor is denoted by $P$, where $(w, c) \in P$, $w, c \in \{1, M\}$. Then, the perpendicularity loss is defined as

$$\mathcal{L}_P = \frac{1}{|P|} \sum_{(w,c) \in P} \cos \angle (p_w, p_c) \tag{4}$$

**Joint loss.** The overall loss is a weighted sum of all loss terms: $\mathcal{L} = \mathcal{L}_T + \alpha_E \mathcal{L}_E + \alpha_P \mathcal{L}_P$.

**Optimization.** We minimize the joint loss with ADAM [23] with a learning rate of 0.1. The plane equations are initialized with ones, and normalized so that the normal direction part is of unit length. The loss is minimized for at most 100k iterations. Optimization is stopped early if there is no improvement for the last 500 iterations. The hyperparameters are set manually to $\alpha_E = 0.1$ and $\alpha_P = 0.1$. The number of samples for point tracking is computed by requiring the average distance between neighboring sampled points to be exactly 30 pixels.

### 3.4   Estimating the 3D spatial extent of planes

At this point, the plane equations of all structural elements are known. As those equations define infinite planes, we now cut them to determine their spatial extent, as defined by the union of all parts that are in the camera field-of-view at any time during the video (even pieces occluded by furniture). We do not extrapolate beyond what the video shows, and so we do not strive for closed room models. We represent spatial extent with a triangular mesh, where each triangle has a label indicating to which structural element it belongs.

**Polygon union.** As the first step, we determine the complete spatial extent of the plane, integrated across all video frames. Usually, no single frame shows the whole extent, with different frames showing different (typically overlapping) pieces. Therefore, we compute the complete spatial extent as the union of extents in each frame. To achieve this, we unproject the polygon annotations of a structural element to 3D using the unproj($\cdot$) function of Sec. 3.3. Since unprojecting 2D polygons from different frames onto the same 3D plane results in coplanar polygons in 3D, we directly compute their union (Fig. 2, bottom box, right).

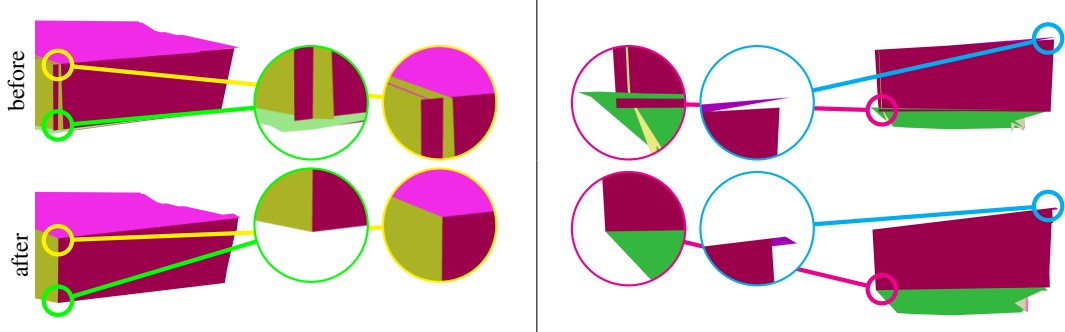

Figure 5: **Spatial extent refinement,** before (top) and after (bottom). We cut hanging walls extending outside the room boundary, and fill in the holes between neighboring planes (blue).

**Refinement.** As shown in Fig. 5 (top), the resulting mesh contains artifacts, *e.g.* hanging wall extending outside the room boundary and small holes between structural elements. Moreover, structural elements never intersect precisely in the 3D space. To address this shortcoming, we introduce a refinement procedure, which ensures that structural elements intersect precisely in the 3D space. We extend each plane's unified polygon towards its neighboring planes until they intersect in 3D. If such extension does not increase the size of the polygon by more than 10%, it is accepted. This procedure fills in the holes between neighboring 3D polygons. Finally, for each pair of intersecting 3D polygons, we find their intersection line and cut the smallest part if it is smaller than 10% of the total area. As shown in Fig. 5, the room layout becomes more visually pleasing after refinement.

**Doors and windows.** Annotators are asked to label all doors and windows separately unless the door is open, in which case the next room must be annotated. We assign each door and window a structural element index, with which it shares most of its border. Then, the door and window segmentation masks are used for spatial extent calculation of their assigned structural elements. Finally, we also calculate spatial extent of each door and window and include them in the final reconstructed mesh.

### 3.5 Automatic quality control

The 3D estimation method in Sec. 3.1 to 3.3 can sometimes fail. This typically happens in the presence of large annotator errors or insufficient camera motion. To ensure high quality, we introduce an automatic quality control mechanism.

We measure Reprojection IoU by rendering the estimated 3D layout with the given camera parameters for each video frame. Such rendering resembles the input 2D structural elements annotation. We compute the similarity between the rendering of each structural element and its annotation as Intersection-over-Union (IoU), and we average over all structural elements and frames in a video (e.g. Fig. 3, comparing structural elements annotations to rendered final layouts).

Since our 3D estimation method is stochastic due to randomly sampled point tracks and stochastic optimization, each run gives different results. We automatically select the best run for each video after $R = 100$ repetitions, based on Reprojection IoU. We then discard all videos with reprojection IoU below 0.8, keeping only high quality reconstructions.

## 4 Experiments

We use our method from Sec. 3 to annotate the RealEstate10k dataset [75] (RE10k), which gathered videos of indoor scenes from YouTube. A total of 21 human raters annotated 3743 of those videos. After keeping only high quality ones (Sec. 3.5), this results in 2246 3D layouts in our dataset. Each video lasts from 3 to 10 seconds, with 6.5 seconds on average, recorded at 30 frames per second. We annotate one frame per second to avoid redundancy, but use all frames for point tracking (Sec. 3.3). Finally, for evaluation purposes, we also annotate 50 scenes from the ScanNet dataset [11].

In the following subsections, we present evaluation measures based on ground-truth 2D segmentations and 3D depth (Sec. 4.1), evaluate the quality of our annotations on ScanNet [11] and RE10k according

| | Runs | RE10k | ScanNet [11] | |
| | | IoU↑ | IoU↑ | $\epsilon \downarrow$ |
|---|---|---|---|---|
| Ours (full method) | 100 | **0.89** | **0.90** | **0.22** |
| Ours (no quality control) | 100 | 0.83 | 0.85 | 0.30 |
| Ours (no quality control) | 30 | 0.81 | 0.84 | 0.33 |
| Ours (no quality control) | 1 | 0.72 | 0.79 | 0.36 |
| Ours (no quality control) | 10 | 0.90 | 0.84 | 0.33 |
| No perpend. loss $\mathcal{L}_P$ | 10 | 0.80 | 0.83 | 0.34 |
| No edge loss $\mathcal{L}_E$ | 10 | 0.72 | 0.84 | 0.33 |
| No points tracking $\mathcal{L}_T$ | 10 | 0.34 | Fail | Fail |

| | RE10k (no depth) | | | ScanNet [11] (depth) | | |
| | Pr%↑ | Rc%↑ | IoU↑ | Pr%↑ | Rc%↑ | IoU↑ |
|---|---|---|---|---|---|---|
| Ours | 88.5 | 98.0 | 0.89 | 83.8 | 91.2 | 0.90 |

Table 2: **Left** Top row: our main results, with our full method. Rows 2-4: ablation removing the automatic quality control filter and reducing the number of runs $R$. Rows 5-8: ablation removing parts of the loss. **Right** Manual inspection on 50 randomly selected scenes from RE10k and 20 from ScanNet. We report precision and recall of whether a structural element is successfully reconstructed.

to those measures (Sec. 4.2), and additionally also by manual inspection (Sec. 4.3). Finally, we evaluate a state-of-the-art 2D layout estimation method [31] on our dataset (Sec. 4.4).

## 4.1 Evaluation measures

**Reprojection IoU.** We first measure quality based on the consistency of our reconstructed layouts with the input 2D annotations, as defined in Sec. 3.5.

**Depth error.** As a second measure, we compute the average absolute depth error $\epsilon$. This is computed only on the visible parts of structural elements, as other parts might contain furniture occluding them. We produce depth maps by rendering our 3D room layout at each frame, and then compare them to the ground-truth depth maps. These are available for ScanNet [11], which was acquired with an RGB-D sensor, but not for RE10k, which contains pure RGB videos.

## 4.2 Results

**Full method on ScanNet.** We validate the quality of our reconstructions on the ScanNet dataset [11] (50 random scenes), based on the ground-truth depth maps released with it. These have been acquired with a structured light scanner and are therefore high quality. As Table 2 (left, top row) shows, the average depth error $\epsilon$ is 0.22 meters. To put this in context: walls are 2.7 meters tall and 3 meters long on average, and rooms in ScanNet have large spatial extent of 7 meters on average. Hence, a depth error of only 0.22 meters represents high quality. This is further confirmed by a very high average Reprojection IoU of 0.90 on ScanNet.

**Full method on RE10k.** Our full method achieves a high Reprojection IoU of 0.89 on RE10k (Table 2, left, top row). This indicates high quality 3D reconstructions recovering most structural elements and their spatial extent correctly.

**Ablation study on RE10k and ScanNet.** Table 2 (left, rows 2-8) show several ablations. In row 2 we turn off the quality control filter of Sec. 3.5. This results is significantly worse Reprojection IoU and depth error, demonstrating that our quality control filter is working well. From row 2 to 4 we gradually reduce the number of runs $R$. Both Reprojection IoU and depth error get continuously worse with fewer and fewer runs. The reconstruction quality is substantially higher at $R = 100$ than for just a single run, demonstrating the value of having multiple runs and of our run selection criterion (Sec. 3.5). Also note how in all these experiment better IoU always corresponds to better depth error on ScanNet, confirming that Reprojection IoU is a good indicator of reconstruction quality.

Finally, we ablate the components of our joint loss (Sec. 3.3) in rows 5-8. The fitting point tracks loss $\mathcal{L}_T$ (2) is the most important one since the optimization fails completely (results in NaNs gradients) on ScanNet scenes (row 8). On the few scenes from RE10k where optimization converged, Reprojection IoU is very low (0.34 on average). Disabling the edge loss $\mathcal{L}_E$ (3) leads to significantly lower IoU

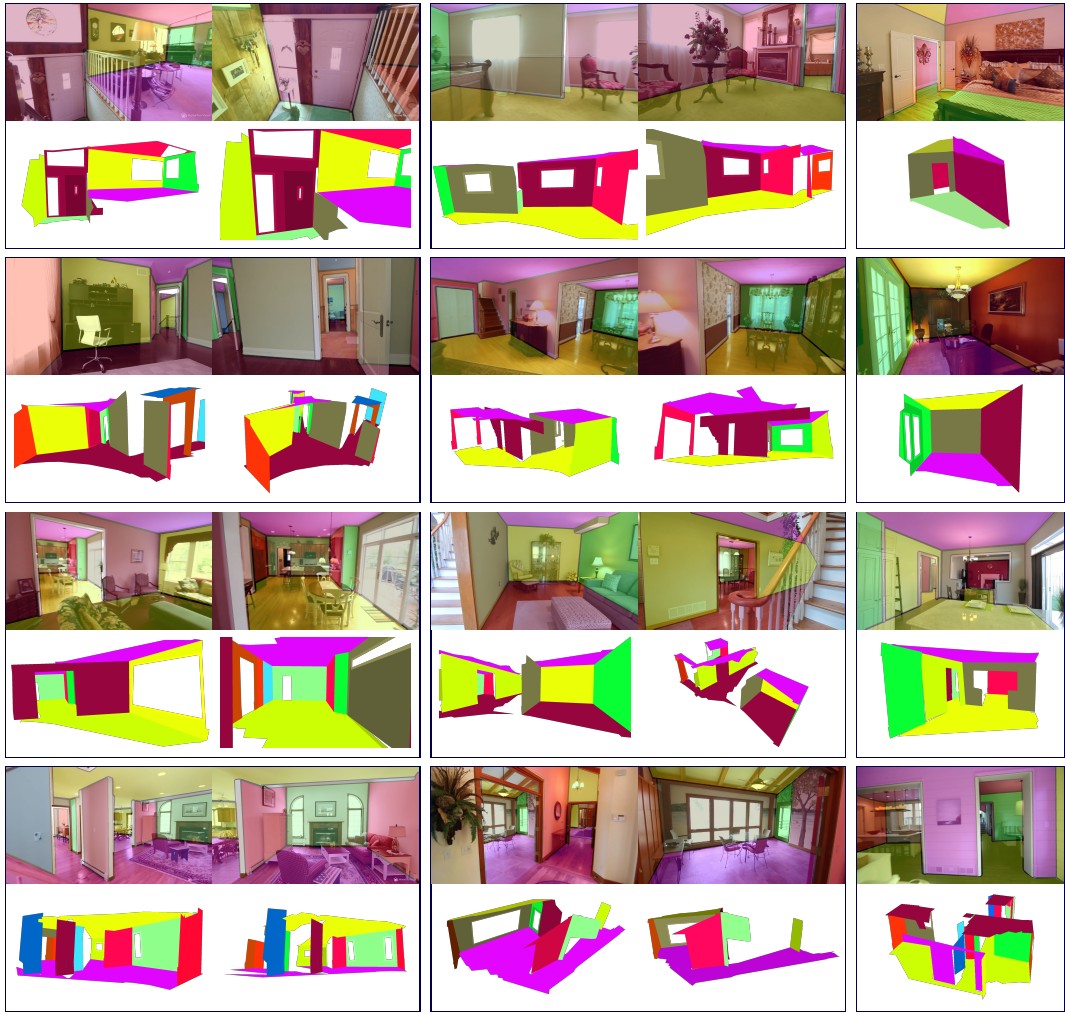

Figure 6: **3D Room layout reconstruction examples.** For each example scene, we show one or two of the input annotations overlaid on the video frame (top), and the final reconstructed layout (bottom).

and larger depth error (row 7), compared to the using the complete loss (row 5). The perpendicularity loss $\mathcal{L}_P$ (4) has a smaller impact on overall performance (row 6).

### 4.3 Manual inspection

We assess here the quality of our 3D layouts by manual inspection of 50 randomly selected scenes from RE10k and 20 from ScanNet. For each scene, we measure (1) the total number of ground-truth structural elements in the scene; (2) the number of elements successfully reconstructed (recall); (3) the number of spurious elements incorrectly created (enabling to measure precision). A structural element is considered successfully reconstructed if the plane normal is correct and the spatial extent is consistent with the 2D annotations (up to a 10% error in both cases). Note that humans cannot verify the correct depth of each plane, as this is perceptually hard to judge. As Table 2 (right) shows, quality is excellent with a recall of 98% and precision of 89% on RE10k.

### 4.4 Automatic layout estimation on our dataset

Most methods for 3D room layout estimation require special inputs, such as *e.g.* RGB-D fusion [40, 4, 8, 18], and therefore cannot be run on our dataset. Some methods input a single RGB image, but they only estimate a 2D room layout (*i.e.* 2D projection of a 3D room layout), *e.g.* [25, 31, 73, 69,

|  | LSUN dataset [66] | Hedau dataset [19] | Our dataset |
|---|---|---|---|
|  | Pixel Error (%)↓ | Pixel Error (%)↓ | Pixel Error (%)↓ |
| Hedau *et al.* [19] | 24.23 | 21.20 | - |
| Mallya *et al.* [35] | 16.71 | 12.83 | - |
| DeLay [12] | 10.63 | 9.73 | - |
| CFILE [45] | 7.57 | 8.67 | - |
| Zhang *et al.* [68] | 6.58 | 12.70 | - |
| ST-PIO [71] | 5.29 | 6.60 | - |
| Lin *et al.* [31] (baseline) | 6.25 | 7.41 | 26.3 |

Table 3: **Baseline method.** We train and evaluate the method [31] on our dataset, following the train and test split. We compare to existing datasets [66, 19] and to other layout estimation methods.

54, 19, 20, 13, 14, 34, 33]. While our dataset offers 3D layout annotations, we evaluate here a 2D estimation method [31] on it. This allows us to get a sense of the difficulty of our dataset and to position it relative to other existing datasets.

We train and evaluate the method [31] on our dataset. To this end, we render our 3D layouts using the camera poses of the video frames, and we run [31] on each frame independently. In Table 3, we report results of [31] on our dataset, and of various methods on two existing datasets [66, 19]. The method [31] performs at the state-of-the-art on the existing datasets, with a low error around $6\% - 7\%$. Instead, it performs much worse on our dataset (26%), demonstrating it offers a harder challenge than the previous ones.

## 5    Limitations

Our 3D room structures only contain surfaces that are observed in the videos. We do not produce geometry for the unobserved parts. For example, if a floor is partially visible behind a wall, we only produce geometry for what we can observe from it in the video. Ideally, we would also like to generate plausible geometry for the unobserved parts.

Another limitation is that we reconstruct only planar surfaces. Our current method cannot handle curved surfaces, which sometimes can occur in upscale apartments.

## 6    Conclusion

We proposed a novel way to annotate 3D room layouts from only 2D annotations. We annotate 2246 general 3D room layouts on a dataset of real-estate RGB videos from YouTube, which we have publicly released. Extensive evaluations confirmed the high quality of this dataset.

**Acknowledgements.**    We thank Prabhanshu Tiwari and Mohd Adil for coordinating the annotation process, and all the annotators who have contributed to creating the dataset.

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
