# OpenReview forum: "Estimating Generic 3D Room Structures from 2D Annotations"
_NeurIPS.cc/2023/Track/Datasets_and_Benchmarks — NeurIPS 2023 Datasets and Benchmarks Poster_

### Official Review · Reviewer_ChJp · 2023-07-06
**The ever-first procedure of creating 3D room layouts from RGB with 2D annotations; procedure + annotations on RE10K**

**Rating:** 8
**Confidence:** 5

**Strengths:**

*First, as I understand, your contribution is two-fold: first, you describe an algorithm for producing 3D layouts from 2D annotations, second, you present a dataset of layouts created by this algorithm. So, I assume that concerns about both the process (algorithm) and the result (dataset) are both legit, and review your work accordingly.*

**Significance.** The contribution is undisputable, considering the fact that there are no other approaches of creating a 3D room layout dataset from RGB videos only. Besides, no other layout dataset shares the same characteristics, so there are no direct competitors to the presented dataset.

**Relevance for a broad community.** This is a quiet niche work, serving for specific purposes, which hardly can be re-conceptualized and used for another tasks rather than 3D layout estimation.

**Quality.** This work seems to be thoughtful and solid, performed at a relatively high technical level, with much attention to details. Overall, it makes a good impression, while I personally find it a bit lacking of motivation behind certain design choices.

**Additional Feedback:**

**Data**. Capturing RGB-D videos require special equipment -- I would argue that the special software is the key, not the equipment. Many modern smartphones carry a depth sensor, like Apple LiDAR, or ToF depth sensors in Samsung devices, so there is no need for a capturing platform, since RGB-D videos can be collected with just a casual smartphone.

**Algorithm**.
1. The loss term penalizing non-straight edges between structural planes, considers only edges between walls and floor/ceiling. Why don't you impose a similar constraint on edges between windows/doors and floor/ceiling? If you do not annotate doors and windows as Slanted, they should be vertical, so it seems natural to calculate loss for doors and windows same as for walls. Maybe you have already tried so, and obtained worse results? Anyway, I would be grateful for any motivation behind this choice, and if this is possible, I would ask for an ablation study.

2. You track points belonging to structural elements across the frames. Do you face any difficulties with tracking failures due to patterned / textureless walls, floor, and ceiling?


**Annotation**.
1. In the Annotation Guide provided in supplementary, you mention Stairs. But in the paper, the only classes considered are Floor, Ceiling, Wall, Slanted (could be a wall, ceiling, etc.), Door, and Window. How the Stairs are processed in your pipeline?

2. How do you process the occlusion edges in general? Besides, by drawing an occlusion edge with a line segment, you implicitly assume that curved shapes are not present. I would easily imagine a window or a door of a shape of an arc (I have actually witness doors being rounded on top in an overall quite casual flat...). How to draw an occlusion edge in this case? If there are no such examples in the datasets you consider in the paper -- how would you handle such a case?

3. Is labeling of structural elements and visible parts performed independently? By a single or several annotators? If this is done by a single annotator, are they asked to maintain spatial consistency between annotation of different types? If multiple assessors are involved, do they have access to the annotations of another types, already performed by other annotators?

3. Do you have multiple annotations for each image, or are images annotated only once? Do you perform any cross-checks, or correct errors in an existing annotation?

4. If only a part of an object protrudes, should this part only be excluded from a layout annotation, or the whole object? Like, for a flat picture in a voluminous frame, should the picture in a frame excluded from a layout, or should the picture be annotated, and the frame ignored?


*You did a good work! I am definitely leaning towards a higher grade. Looking forward to having my concerns addressed in the comments.*

**Clarity:**

The paper is well written and easy to understand. Overall, it was a pleasure to read.

**Correctness:**

The data acquisition procedure seems to be generally correct. There are a number of ablation studies reported. Yet, I have a minor concern on annotation verification.

As I understand, you verify the quality of annotation by calculating depth errors w.r.t. ground truth depth measurements. Since you do not reconstruct the entire scene with all present objects but only 3D layouts, you can only check depth / reprojection error in annotated areas, but cannot validate *whether the proper areas are annotated*. Are annotation issues eliminated by labeling each frame independently, so that individual errors (if any) do not affect the result much? It seems that the more frames are annotated, the higher is the tolerance to labeling errors. Vice versa, if a unique structural element is present in few frames, even a single erroneous mask might be crucial. Here, I base my assumptions on a common sense, but does this really holds in practice? By the way, do you have a statistics of how many times each unique structural element is annotated?

Let's imagine that annotation is erroneous in non-layout areas (some of them are actually layout, but are missing in the annotation). How this kind of errors could be identified non-manually? Maybe you can use auxiliary models for object detection / recognition, semantic or instance segmentation, or even predict surface normals and try to align them with the detected planes? I would love to hear your versions on that, though.

**Documentation:**

The GitHub repo contains all necessary links for downloading the data. Besides, it provides tool for loading and visualizing the downloaded data. The dataset structure is explicated in the README. In supplementary materials, authors describe the intended use, and reveal the hosting and maintenance plan, as well as licensing (also available on GitHub). Moreover, authors share the extremely detailed annotation guide. However, I still have some questions and concerns about the data acquisition protocol (see Additional Feedback).

**Ethics:**

The dataset of indoor spaces contains no images of people, so there should be neither privacy issues nor issues related to social groups representation.

**Limitations:**

1. First, it would be nice to see a 3D layout estimation baseline established using your data -- it seems to be a rule of thumb when presenting a new dataset. At least, I would recommend preparing a training/validation/test split and provide data loaders (e.g., in PyTorch). In Related Work, you mention a number of layout estimation methods, claiming that these methods could benefit from the use of your dataset: while being quite evident, this statement needs though to be supported with an empirical evidence.

2. Next, you mention that only a small part of frames should be annotated to obtain a decent quality. This is a good finding, however, is causes plenty of questions to arise. Have you performed a study to determine the minimal ratio of frames that should be annotated? How the quality of layouts depend on the ratio of annotated frames? Is there a certain threshold, where the quality reaches a plateau and does not improve with the increase of the ratio?

3. In addition, your algorithm requires camera intrinsics and extrinsics alongside annotated RGB inputs; it is claimed that camera parameters might be computed by COLMAP. For ScanNet experiments, do you use (precise) camera parameters provided in the dataset? Or do you assume there are only RGB data is available, and estimate camera poses and intrinsics by yourselves? If you have not conducted such a study, I would definitely advise you doing so, since the scenario with only RGB videos available might highlight the potential and the limitations of your approach in the best possible way.

**Opportunities For Improvement:**

See Limitations.

**Relation To Prior Work:**

The comparison against existing layout datasets is given in Table 1. Overall, the difference is made clear; yet, I would ask to add a column containing number of scans / layouts to judge about the relative scale of data. Besides, I would be grateful for a qualitative comparison (e.g., side-by-side visualizations of layouts provided in various datasets).

**Summary And Contributions:**

This paper introduces a novel method to generate 3D room layouts from 2D segmentation masks annotated for individual RGB frames. Plus, a new dataset of 3D layouts on top of RealEstate10K is presented.

The proposed algorithm requires annotations of two types: 2D instance masks depicting unique structural elements of a scene, and binary masks representing whether each pixel in an RGB image belongs to a visible part of the layout.

Provided with such masks, the proposed algorithm derives 3D plane equations for the elements defining the scene structure, namely walls, ceiling, floor, doors, and windows. The initially obtained planes are post-processed, so that their spatial extent in the scene is restored, and the adjacent planes are connected with edges. The optimization procedure is guided with three losses, enforcing 1) consistency of 3D coordinates of points being observed from multiple views, 2) intersections of 3D planes to be coherent with 2D edges in annotation, and 3) straight angles between walls and floor/ceiling planes.

The validity of the described procedure is demonstrated on the ScanNet benchmark. After that, this procedure was applied to generate 3D room layouts for the RealEstate10k dataset, resulting in 2266 3D room layouts.

The dataset is available on Github; the code for loading and visualizing data structures is provided.

---

> ### Author Response · Authors · 2023-08-18
> **Rebuttal by Authors**
>
> Thank you for your kind words and the detailed and positive review.
> ### **Dataset splits, data loader, and baseline**
> We now provide PyTorch data loaders here:
> https://github.com/google-research/cad-estate/blob/main/src/cad_estate/datasets.py
>
> We provide dataset splits that can be obtained by following the instructions here:
> https://github.com/google-research/cad-estate/blob/main/downloading_the_dataset.md
>
> For the baseline, please refer to the global response.
> ### **Number of annotated frames per scene**
> We annotated every 30th frame to ensure that each structural element is visible at least in two annotated frames. This heavily depends on the camera motion, and our choice is an approximation that worked well. We have now conducted a small user study on scenes that did not pass our automatic quality control filter in Sec. 3.5 (see the answer to reviewer **WKsm**). We found out that the main reasons for failure are annotation mistakes and wrong structural elements association between frames due to full occlusion. We did not find cases where failure could be attributed to the limited number of annotated frames.
> ### **Camera parameters for ScanNet**
> In Sec. 4.1 and 4.2, for the purpose of evaluating the 3D reconstructed layouts produced by our method, we need to compare our depth maps to the ground-truth of ScanNet. This involves computing a depth error in absolute metric space (literally meters). For this, we needed to use the ground-truth camera poses for ScanNet. SfM methods on images without further input cannot produce absolute metric values, which makes a proper comparison of depth maps impossible.
> However, all our results on RealEstate10k are based on camera poses automatically estimated by SfM.
> ### **Annotation mistakes (correctness)**
> We annotate two types of masks on each frame: amodal structure element masks and visible parts masks. We measure depth errors only on annotated parts as visible, as we cannot know what happens behind occluders. Our method is robust to mistakes in the visible-parts annotations. In fact, we only need a non-exhaustive, rough estimate of visible parts that indicate some visible parts. If the annotation mistakes in those visible parts were too large, the final Reprojection IoU would be low, and therefore the scene would be discarded by our automatic quality control (Sec. 3.5). In practice, we never experienced such a case, and therefore we did not need to think about how to automate this with object detection. Most failures we observed were due to the structural-elements annotations (e.g. two perpendicular walls were annotated as one) and not on the visible parts annotation.
>
> Annotating frames independently definitely helps to average out errors in individual frames.
> ### **Statistics of how many times each unique structural element is annotated**
>
> Each structural element is annotated in at least two frames. On average, each structural element is annotated in about 3.5 frames.
> ### **Improvements for Table 1**
> We have included a column with the number of scans and images (in the revised Table 1). In the revised supplementary PDF (Fig. A2), we provide an example layout for each dataset for qualitative comparison.
> ### **Constraints for windows/doors**
> Each door or window is assigned to the structural element with which it shares most of its boundary (**L204-205**). This means that a door/window cannot exist on its own without being part of some other structural element and therefore does not have a separate 3D plane equation. In most cases, it will share a 3D plane equation with a wall.
> ### **Point tracking difficulties**
> We use RAFT optical flow for point tracking. It is known to be robust to patterned and even textureless surfaces. Moreover, it is rare for a structural element to be completely textureless and uniformly colored. We did not experience difficulties with point-tracking precision.
> ### **Annotating stairs**
> We did annotate the stairs. However, we did not find a useful way to process or exploit them. We thus ignore them in the current reconstruction pipeline.
> ### **Occlusion edges**
> Occlusion edges are not constrained to straight lines. In fact, they were annotated with arbitrary curves using the mouse. We apologize for this confusion and already clarified this in the revised paper (**L113**).
> ### **Independent labeling**
> The labeling of structural elements and visible parts is performed independently. Moreover, the labeling of each frame in each of these two types of annotations is also performed independently, and they can be parallelized. Both tasks were performed by several annotators. Annotators do not have access to the annotation of the other type.
> ### **Multiple annotations**
> Each image is annotated only once, but an expert double-checks the answers of both tasks and provides feedback to the annotators.
> ### **Partially protruding objects**
> We could both ignore or annotate parts of such objects since we only require at least some regions for tracking.

---

### Official Review · Reviewer_ujhv · 2023-07-18
**The proposed method can automatically generate 3D layouts from 2D video annotations.**

**Rating:** 6
**Confidence:** 3
**Correctness:** The claims are correct. There is no w…
**Clarity:** The paper is clear.

**Strengths:**

- The proposed method is the first for machine-assisted 3D room layout annotation. They can generate the 3D layouts of indoor scenes using 2D RGB videos, without 3D data.

- The method can automatically generate 3D layouts from 2D annotations of 2D structural elements annotation and 2D visible parts annotation.

- The experiments on the reprojection of 3D layouts to 2D validate the reconstruction accuracy of proposed datasets.

**Additional Feedback:**

nil

**Documentation:**

The authors provide the document to generate 3D annotations. There is no link to download the final dataset directly.

**Ethics:**

There are no or only very minor ethics concerns.

**Limitations:**

There is no limitation section in the paper.

**Opportunities For Improvement:**

- The motivation of the proposed datasets needs further discussions.

- The experiments only include the validation of the constructed datasets. The authors may provide a downstream task benchmark for the proposed dataset, which can better illustrate the value of the dataset.

**Relation To Prior Work:**

The paper discusses the relation to previous 3D room datasets in Sec. 2.

**Summary And Contributions:**

This paper proposes a new method to produce generic 3D room layouts from 2D segmentation masks automatically. Based on the method, the authors utilize annotations by humans on 2D RGB video to reconstruct 3D plane equations for the structural elements and their spatial extent in the scene. They annotate and publicly release 2266 3D room layouts on the RealEstate10k dataset.

---

> ### Author Response · Authors · 2023-08-18
> **Rebuttal by Authors**
>
> Thank you for your valuable review.
>
> ### **”There is no link to download the dataset”**
>
> We released the full dataset as per official NeurIP’23 dataset track rules at submission time (https://github.com/google-research/cad-estate). Reviewers **ChJp** and **42Nb** have commented positively on the fact that **the dataset is publicly available**.
>
>
> ### **Motivation for the proposed dataset**
>
> Please refer to the global response about our motivation, and the differences to existing datasets.
>
> ### **Downstream task benchmark**
>
> Please see the global response above.
>
>
> ### **”There is no limitation section in the paper”**
>
> Please see the global response above.

---

> > ### Comment · Reviewer_ujhv · 2023-08-29
> > **Thanks for the rebuttal**
> >
> > Dear authors,
> >
> > Thanks for your rebuttal. After reading the response and other reviews, I believe it is an interesting work for analyzing 3D room structures. I will keep my rating.
> >
> > Best,
> >
> > Reviewer ujhv

---

### Official Review · Reviewer_aGuE · 2023-07-22
**Paper Review**

**Rating:** 6
**Confidence:** 4
**Correctness:** Yes.
**Clarity:** Yes.

**Strengths:**

- This is a novel and smart way to annotate 3D scene layout. Many researchers have tried to annotate 3D from 2D annotations, but the proposed approach finds a good perspective.
- The annotation quality is high, which is validated by humans and on ScanNet. This is critical to a 3D dataset from 2D annotations.

**Additional Feedback:**

N/A

**Documentation:**

Yes.

**Ethics:**

No. They work on realestate10k, which is a public dataset.

**Limitations:**

Yes in the supplemental.

**Opportunities For Improvement:**

My primary concern is: whether it is really better and more efficient than previous methods.

First, I try to understand the difference between the proposed and previous datasets. The related work has discussed the comparison with previous datasets (e.g. Zillow indoor datasets). But I do not completely understand the advantage. I do understand the proposed approach can construct generic layouts, which is clearly better than Cuboid or Manhattan assumptions. And I think it's beneficial to move beyond the Manhattan world assumption. However,

 > (L67) Some datasets [9, 4] construct generic room layouts (like ours), but are limited by the data capturing process (see above).

> (L60-62) A few real datasets with 3D room layouts were introduced, but they heavily rely on special acquisition devices or sensors to create the 3D annotation of the scene, which limits the range of scenes they can cover.

Can you elaborate what [4, 9] is limited? My understanding is, the only advantage is they need special device and the proposed approach only needs a video, which is more easily accessible. Let me know if it's wrong or there are other advantages.

Second, assume the only advantage is it only needs a video, I think the advantage is not fully validated. To show the proposed approach is more effective, can you show

- How much does it cost on average to annotate a single scene?
- How long does it take on average to annotate a single scene?

Third, I think the advantage is not fully utilized. Technically it can be run on any videos, but eventually only 2,266 scene layouts are generated. I don't think this is a large number. For reference, MatterportLayout (although it has Manhattan assumption) has 2,295 layouts. If we want to fully utilize the advantage, I expect it to be run on multiple sources of videos and the final datasets should be larger than any previous datasets.


**Relation To Prior Work:**

Yes.

**Summary And Contributions:**

The paper proposes a novel way to annotate 3D room layout based on video frames and 2D annotations. Using the proposed approach, it is easier to annotate high-quality 3D room layout. The approach is tested on realestate10k dataset and generates 2,266 scene layouts. The full method is further validated by humans and ScanNet to verify the quality.

---

> ### Author Response · Authors · 2023-08-18
> **Rebuttal by Authors**
>
> Thank you for your valuable review.
>
> ### **Contributions and advantages wrt. SceneCAD [4] and Zillow [9]**
>
> Please see the global response above.
>
>
> ### **”Dataset size (2266 scenes) is not large enough”**
>
> We provide a dataset that is as large as Matterport3D and larger than Zillow (see global response above). Our main contribution is the qualitative difference between those datasets: our dataset features arbitrary room geometry, unconstrained camera poses, and furnished rooms. This is enabled by our new method for annotating any video in 2D and converting these annotations into 3D layouts.
>
>
> ### **Annotation times**
>
> On average, and for a trained annotator, it takes 10 minutes per scene to annotate structural elements and 5 minutes for visible parts, so 15 minutes in total. We emphasize that other works need a significant amount of acquisition time and cost (hardware, commute, personnel), whereas we simply download the videos from Youtube.

---

> > ### Comment · Reviewer_aGuE · 2023-08-26
> >
> > Thank you for the response. In this case, I confirm I see value of this paper. I'll keep my rating.

---

### Official Review · Reviewer_WKsm · 2023-07-24

**Rating:** 6
**Confidence:** 5
**Correctness:** Sound method.
**Clarity:** Clear to me.

**Strengths:**

+ The method is sound. I have been thinking for a while about the same idea: using pytorch optimizers to get 3D planes on multiview images.
+ A large-scale evaluation demonstrates that the metholody's scalability.
+ This paper comes from Google and Niesser's group, which is not a technical issue but potentially an advantage. This may (re-)trigger interests in this important field that I've been working on for years.

**Additional Feedback:**

None.

**Documentation:**

Not released yet.

**Ethics:**

None.

**Limitations:**

None.

**Opportunities For Improvement:**

+ I am quite interested in the failure cases that are filtered out during quality control. The sample number decreases from 3743 to 2266 and I would like to see how they fail as I believe they reveal the limitations of the method.
+ The authors only claim to release the dataset. However, the annotation protocol itself is also an interesting interactive room layout estimation method. I believe for NeurIPS datasets track, the tool should also be released.
+ The authors claim 0.22 meters error in an indoor scene is low. I can only partially agree. I have been trying to sell my room layout estimation algorithms to the industry for years and the feedback is always that 5cm error is still to high for construction and design, let alone 22 cm.
+ Missing references to [A], which is also an optimization-based room layout estimator and [B][C] which are state-of-the-art room layout estimators using 3D points.

[A] 3D room layout estimation from a single RGB image, TMM 2020

[B] Pq-transformer: Jointly parsing 3d objects and layouts from point clouds, ICRA 2022

[C] From semi-supervised to omni-supervised room layout estimation using point clouds, ICRA 2023

**Relation To Prior Work:**

Missing references.

**Summary And Contributions:**

This submission proposes a new optimization-based methodology to annotate room layouts from RGB videos. The annotation involves amodal 2D layout masks (for instance identificaiton) and visible points on them (for keypoint tracking and 3D reconstruction). The core idea is using modern deep learning random optimizers (ADAM) to find optimal 3D plane equations that allow backprojected point tracks to joint at a single 3D point. Other two losses that enforce the consistency between 3D edges and 2D edges and the orthogonal constraints are also exploited. After 3D plane equations are reconstructed, they are cut to form polygons using rules. Evaluations on RE and ScanNet show good reprojection consistency and depth accuracy.

---

> ### Author Response · Authors · 2023-08-18
> **Rebuttal by Authors**
>
> Thank you for your valuable review.
>
> ### **Failure cases**
>
> We discussed failure cases and limitations in the supplementary material (Sections **2** and **3**).
> As mentioned there, failure cases occur from annotation mistakes (wrong labels, inaccurate structural element boundaries, missing structural elements). This can lead to the following failure cases: wrong plane orientations, poor spatial extent estimation, and duplicate structural elements. Importantly, our automatic quality control mechanism filters out these cases (Sec. **3.5**). These removed cases are not released as part of the dataset.
>
> We now additionally conducted a small user study by randomly sampling 50 scenes that were filtered out by the quality control and identify why they failed:
> - About 30% failed because of annotation mistakes (wrong labels).
> - About 40% failed because of incorrect association of structural element masks across frames, which leads to duplicate structural elements in the final reconstruction. This usually happens when the same structural element reappears after full occlusion.
> - Another 30% failed because of degenerate cases (e.g. very short sequences with purely forward camera motion, or very few tracked points on structural elements due to massive clutter and furniture occluding them, or very small structural elements).
>
> We included a new Fig. **A1** and a new failure case section (Section **2**) in the revised supplementary PDF.
>
>
> ### **Releasing the annotation tool**
>
> We will release the software to generate our 3D room layouts from 2D annotations.
>
>
> ### **On the depth error**
>
> Our error of 0.22m (in 7m wide rooms) might be high for construction and design, but it is low enough for other applications, especially for training and evaluating computer vision methods for automatic 3D reconstruction.
>
>
> ### **Missing references**
>
> We have included and discussed the missing references in the revised paper (**L79-81**).

---

> > ### Comment · Reviewer_WKsm · 2023-08-20
> >
> > I have raised the score to BA. I like the failure cases analysis. It reveals that room layout estimation is still far from solved despite an interactive annotator with multiple constraints is used. Hope this new dataset could lead us to the ultimate goal of truly useful room layout estimation.

---

### Official Review · Reviewer_42Nb · 2023-07-27
**Good work, but unclear advantages with respect to other existing datasets**

**Rating:** 6
**Confidence:** 4
**Clarity:** The paper is in general well written.

**Strengths:**

* The method for generating the ground truth is reasonable and well explained.
* The ablation of the method is solid and evidences that, although not perfect, the method offers high quality layouts.

**Additional Feedback:**

Nothing to add, looking forward to reading the responses of the authors to my comments.

**Correctness:**

The method developed seems correct to me. The qualitative results in Figure 6 suggest that, at least for these examples, the dataset was created in a sound way.

**Documentation:**

The documentation is clear and the data is already on a public website. The license and maintenance plans seem reasonable.

**Ethics:**

I do not foresee any ethica problem with the data in this dataset.

**Limitations:**

The authors do not discuss the potential limitations of their dataset in the paper. As I pointed out in the previous section, in my opinion, the dataset created migh have some weaknesses.

The authors did not comment in societal negative impacts. As far as I checked, I think their work does not have a societal impact.

**Opportunities For Improvement:**

* As the main weakness of the work, I think that the contribution with respect to existing datasets is unclear from the paper. Table 1 references several datasets of the literature that includes layout ground truth for generic room structures. The only difference is that the referenced ones are for panoramic images, however, images with variable fields of view can be generated from panoramic images. If I understood the dataset specs correctly, Zillow Indoor has a significantly larger of images, which is a technical spec that is not reported for other datasets in the literature. Finally, the paper does not validate the dataset quantitatively with experiments for the layout estimation task. As a result, the dataset could be too complex or too easy for current methods, and hence of little use compared to others in the literature. Although the main selling point of the paper is the novelty of the method to generate the layouts, it is unclear what the data generated with such method is adding to current datasets and if it is going to have the correct level of complexity to be useful to researchers in layout estimation.

**Relation To Prior Work:**

Up to my knowledge, the related work section, although maybe not exhaustive, cites the main works related to this dataset.

**Summary And Contributions:**

The authors created a dataset for 3D generic (not necessarily cuboid or Manhattan) layout estimation. The ground truth layouts are generated with a novel method that leverages geometric constraints and human annotations of the layout structures in the 2D images. The quality of the generated layout ground truth is assessed via consistency with the 2D manual annotations and depth error.

---

> ### Author Response · Authors · 2023-08-18
> **Rebuttal by Authors**
>
> Thank you for your valuable review.
>
> ### **Contribution with respect to existing datasets such as Zillow Indoor**
>
> Please see the global response above.
>
> ### **Baseline for layout estimation**
>
> Please see the global response above.
>
> ### **”There is no way to judge the complexity of the dataset”**
>
> We intended to make our dataset as complex as the real world without constraints. We annotate videos of scenes with arbitrary geometry, some even composed of multiple rooms connected by doors or staircases, without imposing any of the constraints that previous works do (discussed in the comparison to Zillow in the global response). We, in fact, succeeded in annotating YouTube videos of homes that were not captured for the purpose of reconstruction but rather for selling a property. Given that our dataset reflects the real world without limitations, we believe that it raises the bar and will drive the field forward.
>
> ### **”Potential limitations are not discussed”**
>
> Please see the global response above.

---

> > ### Comment · Reviewer_42Nb · 2023-08-25
> > **The paper improved with the additions and authors' replies addressed nicely my main concerns**
> >
> > Dear authors,
> >
> > First of all, thank you for your detailed responses to my comments and your additional work to address them.
> >
> > Based on the new information during the rebuttal, I will raise my rating to marginally above acceptance threshold. Specifically:
> >
> > 1) The new Table 3 in the revised version of the paper shows that the dataset is considerably more challenging than others, at least for the model of Lin et al. At the same time, it is not extremely challenging so that methods cannot be properly evaluated, and it is in the sweet spot where datasets are useful. I encourage the authors to run more baseline methods in their data, it would be nice to have more numbers to be sure of the usefulness of the dataset, and have a more solid initial baseline in the task.
> >
> > 2) The contributions over existing datasets are properly argued in the general comments to all reviewers. I am particularly convinced by the more general recording setup, sorry if I missed that point when I first reviewed the paper. In any case, I recommend to the authors that they highlight more in the paper these novel points with respect to other data.
> >
> > 3) The extended limitations section is now more comprehensive and better.

---

### Author Response · Authors · 2023-08-18
**Global Response**

We thank the reviewers for their valuable feedback. We are glad that they found our method well explained (**42Nb**) and scalable (**WKsm**), our ablations solid (**42Nb**), and our contributions undisputable (**ChJp**). We provide a novel way (**aGuE**, **ChJp**)  to generate 3D layouts of indoor scenes using 2D RGB videos without 3D data (**ujhv**, **ChJp**). We released the data publicly, with clear documentation (**42Nb**, **ChJp**).
In this global response, we addressed issues raised by multiple reviewers. We also include the revised paper and supplementary materials and mark our changes in blue color.

### **Contribution wrt. other datasets Zillow [9], MatterportLayout [69], and SceneCAD [4]**

#### **Contents of the dataset (rev. 42Nb, aGuE, ujhv)**

Existing datasets, including Zillow [9] and MatterportLayout [69], provide layouts in a controlled setting. Zillow’s layouts assume an Atlanta world (horizontal floor and ceiling, vertical walls) and unfurnished rooms. The cameras are always at the same height because of their specific acquisition process. MatterportLayout [69] assumes a Manhattan world, which is even more constraining. Instead, we provide layouts of arbitrary geometry and support furnished rooms. The camera poses in our dataset are more realistic and natural, as we place no constraints: we use readily available YouTube videos that were filmed for selling houses. The nature of the raw data is also different, as we provide videos instead of panorama images.

In terms of quantity, our dataset compares favorably to Zillow. We did not report the number of images in the submission because we annotated videos. In fact, we provide more images (with their poses) than Zillow: Zillow contains 71,474 panoramas in 1,524 homes, whereas we provide >300k video frames in 2266 scenes. MatterportLayout provides a similar number of scenes to us (2266 vs. 2295) but with a Manhattan world assumption. These findings are summarized in the revised Table **1** in the main paper.

#### **How the dataset is constructed (rev. aGuE, 42Nb, ujhv)**

Zillow [4] and MatterportLayout [9] require panoramic images captured with a very specific setup. SceneCAD [4] annotates on RGB-D scans. Their capturing and annotation processes require the following:

- Specialized equipment (e.g. 360 deg. camera paired with an iPhone for Zillow, Kinect for SceneCAD).
- A person needs to physically access the scene to film it with that equipment, which is very costly.
- Specific capturing instructions (e.g. images captured on a tripod in specific locations in Zillow, zoomed-in views on objects required for reconstruction in SceneCAD).
- After data acquisition, annotating the dataset requires several difficult manual steps. SceneCAD [4] requires manual work for correcting errors in the plane extraction, manually verifies which 3D plane proposals in RANSAC are plausible, and manually refines 3D layouts with Blender. Similarly, for Zillow [9], cleanup and verification processes are manual.

Our dataset construction process does not involve any of the above compromises, making it more general and easy to use. We do not need to be physically present at the scene. We do not need specialized hardware or specialized acquisition instructions. The only manual part of our annotation process simply requires annotating surfaces in 2D, and our verification process is automatic. Finally, we emphasize that we annotate natural videos from the internet that were not captured for the purpose of 3D reconstruction (but for selling the properties), featuring arbitrary camera drives and room geometry.

### **Baseline methods (rev. 42Nb, ujhv, ChJp)**

During the rebuttal period, we trained and evaluated a baseline method [A] for 2D room layout estimation, using their official software release [B]. The results are reported in the revised paper submitted with this rebuttal (Sec. **5**). These results demonstrate that our dataset offers a harder challenge than the previous dataset for this task. We would like to stress that current methods cannot fully leverage our data, as the only known method to predict a 3D room layout from RGB inputs is [C] (with Manhattan room assumption, suggested by reviewer **WKsm**), but their implementation is not released. Existing 3D layout estimation methods are designed to work with 360 images, panoramas, and depth inputs. We believe that our data will push the field forward to design methods that can predict 3D room layouts given only RGB inputs.

[A] Lin et al, "Indoor Scene Layout Estimation from a Single Image," ICPR 2018

[B] https://github.com/leVirve/lsun-room

[C] 3D room layout estimation from a single RGB image, TMM 2020

### **Missed limitations section (rev. 42Nb, ​​ujhv)**

We, in fact, discuss limitations in the supplementary material (Section **2**, **L22** - in the initially submitted version, not the revised one). We extended the limitation section in the revised supplementary PDF and included the failure cases.

---

> ### Comment · Area_Chair_VqCc · 2023-08-29
>
> Dear Reviewer ujhv and ChJp,
>
> We are approaching to the end of the author-reviewer discussion period. We kindly request that you review the authors' rebuttal and indicate whether your concerns have been satisfactorily addressed. If your concerns are well addressed, please also update your scores accordingly.
>
> Thank you for your valuable contributions to this process.

---

> > ### Author Response · Authors · 2023-08-31
> >
> > Dear Reviewers and Area Chair,
> >
> > With the discussion period coming to an end, we would like to thank you for your insightful feedback and for helping us improve the quality of the paper. We are glad that all reviewers now unanimously support our paper for acceptance.
> >
> > --The authors

---

### Decision · Program_Chairs · 2023-09-22

**Decision:**

Accept (Poster)

**Comment:**

This paper received overall positive reviews. Reviewers generally regard the method for generating the ground truth 3D room structures novel and sound and the ablation study solid. They initially raised some concerns: dataset motivation, how this dataset compares with other dataset like Zillow Indoor, failure case analysis, release of the annotation tool, algorithm advantage, etc. The rebuttal was persuasive and addressed most concerns. The final ratings unanimously recommend acceptance. The AC checked the paper, rebuttal, and review comments, and recommends accepting the paper.